# Performance Outcome Measures in Padel: A Scoping Review

**DOI:** 10.3390/ijerph19074395

**Published:** 2022-04-06

**Authors:** Alejandro García-Giménez, Francisco Pradas de la Fuente, Carlos Castellar Otín, Luis Carrasco Páez

**Affiliations:** 1ENFYRED Research Group, Faculty of Health and Sports Sciences, University of Zaragoza, 22001 Huesca, Spain; 629251@unizar.es (A.G.-G.); castella@unizar.es (C.C.O.); 2Department of Physical Education and Sport, University of Seville, 41013 Seville, Spain; lcarrasco@us.es

**Keywords:** padel, anthropometry, physiology, physical performance, biomechanic, injury, match analysis

## Abstract

Padel is a modern doubles racket sport which has become popular around the world in the last decades. There has been an increase in the quantity of scientific research about this sport in the last years. Therefore, the main objective of this scoping review is to provide an updated contextualization of research regarding padel. PRISMA ScR was used in order to search for articles fulfilling the inclusion criteria in five fields of interest: the anthropometric profile, physiology and physical performance, biomechanics, the epidemiology of injuries, and match analyses Seventy-seven records were included in the study. Padel is an emerging sport both in sport and research terms. This scoping review provides coaches and researchers with all the knowledge available in the five fields of interest. Furthermore, this study enables them to make a map of the current state of the research about padel, and it opens up doors to future investigations.

## 1. Introduction

Padel is a modern doubles racket sport invented in Acapulco (Mexico) in 1962 [1]. After becoming very popular in Spain in the last decades, with more than 4 million regular practitioners, and positioned in the top 10 most practiced sports in the country [2], this sport has an international presence in 44 countries around the world [3]. Today, world-wide professional tournaments are celebrated in Spain, Portugal, Mexico, Argentina, Qatar, Italy, United Arab Emirates, United States, Sweden, Germany, Ecuador, Uruguay, Holland, Paraguay, Lithuania, Chile, Ireland, Japan, Denmark, and France [4].

In padel, two pairs of players confront each other following a tennis scoring system and similar rules regarding the timing, position of players, sides, service, returns, and scoring, with the difference being that it is played in an enclosed synthetic glass and metal court allowing the ball to bounce on lateral and back walls for rallies [5]. The court measures 20 m × 10 m (length × width) and it is divided by a normal tennis net (0.88 m at the center strap and 0.92 m at the post) in the middle [5]. The back wall is 3 m × 10 m (height × length) and the side walls are 3 m × 2 m, ending on another 2 m × 2 m wall [5]. The rest of the court, for each half and the later side, consists of two metallic panels of equal dimensions (3 m × 2.59 m) and one gate (2 m × 0.82 m) [5]. Each half court is composed of two service boxes, defined by the service line, which is parallel, and at 6.95 m distance with respect to the net and a perpendicular line dividing the service and net lines [5]. Hence, these characteristics—where players have a theoretical responsibility area of 5 m × 2.5 m, as well as the walls around the court—let them lengthen rallies in comparison to other racket sports, such as badminton and tennis, with bigger courts and no walls [6].

In the last years, an augment in scientific research around padel has arisen to a better understanding of its characteristics and requirements both for professional and non-professional players (Figure 1). Therefore, some studies have investigated its specific anthropometrics [7,8], biomechanics [9], epidemiology [10,11], physiological and physical requirements [12], temporal structure, tactics, and strokes [13,14,15,16]. Otherwise, there have been few studies aimed at reviewing the data available concerning padel [17,18,19] with the purpose of a better global understanding of this sport. This might probably be due to the lack of research over the last years on the traditionally-called “major racket sports” [20], such as tennis [21,22], table tennis [23], badminton [24], and squash [25].

With this background, the objective of this scoping review was to provide an updated contextualization of the sport of padel regarding its anthropometrics, physiological and physical demands, biomechanics, epidemiology, and match analyses to draw future research directions to a better understanding of the sport and to bring scientific knowledge closer to both coaches and athletes to enhance performance.

## 2. Materials and Methods

### 2.1. Search Strategy

Data was obtained following the PRISMA ScR method [26] in order to identify articles published before 31 January 2022 in the five fields of research for the study: anthropometrics, physiology and physical performance, biomechanics, epidemiology of injuries, and match analyses. The PubMed, Scopus, and SPORTDiscus databases were searched for relevant articles. The search strategy in PubMed used the following search terms:For the anthropometric profile: (“padel” OR “paddle tennis”) [Title/Abstract] AND (“anthropom*” [Title/Abstract] OR “body composition” [Title/Abstract])For the physiology and physical performance: (“padel” OR “paddle tennis”) [Title/Abstract] AND (“physiolog*” [Title/Abstract] OR “athletic performance” [Title/Abstract] OR “exercise test” [Title/Abstract] OR “metabolism” [Title/Abstract] OR “aerobic” [Title/Abstract] OR “anaerobic” [Title/Abstract] OR “oxigen consumption” [Title/Abstract] OR “biochemical” [Title/Abstract] OR “haematol*” [Title/Abstract])For the biomechanics: (“padel” OR “paddle tennis”) [Title/Abstract] AND (“biomechanic*” [Title/Abstract] OR “kinematic*” [Title/Abstract] OR “kinetic*” [Title/Abstract] OR “dynamic*” [Title/Abstract] OR “angle” [Title/Abstract] OR “speed” [Title/Abstract] OR “rotation” [Title/Abstract] OR “moment” [Title/Abstract] OR “force” [Title/Abstract] OR “forehand” [Title/Abstract] OR “backhand” [Title/Abstract])For the epidemiology of injuries: (“padel” OR “paddle tennis”) [Title/Abstract] AND (“injur*” [Title/Abstract] OR “illness” [Title/Abstract] OR “pathology” [Title/Abstract] OR “disease” [Title/Abstract] OR “epidemiology” [Title/Abstract])For the match analyses: (“padel” OR “paddle tennis”) [Title/Abstract] AND (“game” [Title/Abstract] OR “match analysis” [Title/Abstract] OR “total time” [Title/Abstract] OR “stroke” [Title/Abstract] OR “time-motion” [Title/Abstract])

### 2.2. Inclusion Criteria

Only full-text articles written in English or Spanish and published before January 2022 were included in this study. Only able-bodied players were considered, regardless of their level (amateur to elite). The inclusion criteria for each field were articles regarding these topics:Anthropometric profile: body composition, morphology, and anthropometric measurements;Physiology and physical performance: physiological responses, energetic demand, aerobic and anaerobic metabolism, strength, and power;Biomechanics: kinematics and kinetics of the padel player;Epidemiology of injuries: acute and chronic pain and traumatology during practice;Match analysis: temporal, game and/or stroke analysis, hand dominance, and situational variables.

Review articles and articles focused on padel instructions or racket/ball analyses were excluded.

The study selection was performed by two independent reviewers, which would avoid abusively eliminating an article, based on the abstracts and keywords. A data-charting form was jointly developed by the reviewers which was continuously updated in an interactive process. Discrepancies among reviewers were discussed involving a third author until they reached a consensus. Data was abstracted based on article characteristics (year, authors, and title) and grouped according to the main focus (anthropometric profile, physiology and physical performance, biomechanics, injury, and match analyses).

## 3. Results

The initial search returned 483 studies in database searching and five studies via other sources—two via citation searching and three reported by the authors. After the removal of duplicates, we screened 156 records, from which 72 articles were reviewed and included based on the inclusion and exclusion criteria (Figure 2). Included sources of evidence are synthesized in Table 1.

### 3.1. Anthropometric Profile

Regarding the six articles related to padel anthropometrics [7,35,56,62,63], men showed higher values in body weight, height, and body mass index (BMI) and a lower body fat percentage than women when comparing national and world-class level players [35,62]. Contrary to this, Martínez-Rodríguez et al. [56] found higher body fat percentage values in college male players compared to those reported by Pradas et al. [62,63] in elite female players. The predominant anthropometric profile both in male and female players is meso-endomorphic when playing at high level [8,35,56,62,63] and a predominantly endomorphic profile was found at lower competition levels [7]. These anthropometric differences among playing levels may be due to the greater amount of training hours invested by national and world-class players.

### 3.2. Physiology and Physical Performance

Laboratory tests show that, in padel players, the VO_2max_ is in a range between 38.4 ± 0.7 mL/kg/min and 55.64 ± 8.84 mL/kg/min depending on gender and level. In addition, their first (VT_1_) and second ventilatory thresholds (VT_2_) are placed between 72%VO_2max_ and 84–85%, respectively [31,46,53].

During matches, Carrasco et al. [31] and Hoyo et al. (2007) [53] found VO_2_ values below 50%VO_2max_. The maximum and mean heart rate follows a similar pattern both in top-level and amateur padel players, where some authors reported values between 154–179 bpm and 130-151 bpm, respectively. When comparing those values to the absolute maximum heart rate, they correspond to 80–85% of the maximum heart rate (HR_max_) and 68–74%HR_max_ for the mean heart rate. In their study, Díaz-García et al. [46] found that amateur players stayed 97.75% of the time below the aerobic threshold and 2.25% between VT_1_ and VT_2_ which confirms, at the cardiovascular level, that amateur padel yields predominantly aerobic efforts.

Lactate concentration seems to follow a similar stable pattern during games, starting below VT_1_ in the beginning (1.83–1.90 mmol/L) and reaching values between VT_1_ and VT_2_ (2.40–3.38 mmol/L) [12,28,63]. Regarding the rate of perceived exertion, it appears to be similar (*p* < 0.05) after the first set and the end of the game [28], wich reinforces the previously described aerobic stable pattern followed by VO_2max_, HR and lactate.

When comparing different competitive levels, Castillo-Rodríguez et al. [12] found a significantly lower cardiovascular response (*p* < 0.05) in high, than in middle and lower level players, shown as a higher time spent between 50–70% of maximum heart rate (43.7% vs. 15.2%), a lower amount of time between 80–90% (12.9% vs. 32.8–30.4%), and a lower RPE, suggesting a better cardiovascular fitness at better levels. Equally, but with no statistical significance, maximum lactate values were lower in high (2.87 mmol/L) than in medium (2.74 mmol/L) and low (3.38 mmol/L) level players.

#### 3.2.1. Physical Performance

Various studies have researched physical performance in padel through different standardized tests [8,42,64,67,87]. Significant differences (*p* < 0.05) have been found between sexes in distances covered during YOYO IR1, vertical jump (VJ), squat jump (SJ), countermovement jump (CMJ), Abalakov jump (ABK), medicine ball throwing (MBT), maximum handgrip isometric strength (MHIS), resistance handgrip isometric strength (RHIS), 10 m sprint, and 20 m sprint [58,65,67,87].

When comparing different amateur levels, lower HRs after YOYO IR1 have been reported for high level players both in men and women (*p* < 0.05), which could reflect their better cardiovascular fitness. On the other hand, high levels of men’s values were lower than medium and low level players in the distance covered during YOYO IR1, MBT, and maximum HIS. The authors hypothesized that high level players would have better technical and tactical development to compensate for this limitation. As for women, high level players covered more distance during YOYO IR1 and jumped higher in VJ than medium level players (*p* < 0.05), showing better cardiovascular fitness and lower body strength [58]. Courel-Ibáñez and Herrera-Gálvez [42] found in high level players a better groundstroke accuracy (*p* < 0.05) and a better heart rate recovery after 60 s (*p* < 0.05) than low level players. On the other hand, they reported worse anterior and posteromedial balance (*p* < 0.05), suggesting a high potential injury risk. As for elite and sub-elite players, a better lumbar isometric strength (*p* < 0.05) has been found for elite players [8].

Finally, Courel-Ibañez et al. [41] reported better fitness conditioning in adult women (35–55 years old) padel players than in a sedentary group, shown as better results in abdominal endurance, vertical jumps, one-foot balancing, and cardiovascular fitness (*p* < 0.001). This could serve as a point of departure for future research in physical activity interventions through padel in this age group.

#### 3.2.2. Mental Performance

Díaz-García et al. [45] observed an increase in mental fatigue from pre- to post-world padel tour matches (*p* < 0.01) and an impairment in reaction time (*p* < 0.04).

#### 3.2.3. Haematology and Biochemistry

Bartolome et al. [29] studied the responses of seven trace minerals to a competition match in high level male players. They found a significant increase (*p* < 0.05) in Cu, Ni, and Zn values, suggesting an increase in the antioxidant and energetic demands as a consequence of catabolism after an acute effort involving frequent mechanical impact due to jumps, turns, and explosive actions. On the other hand, they observed a decrease (*p* < 0.05) in Li values, which may indicate a biological redemption process in the organism to avoid its lack and to ensure an optimal development of several biological systems, as well as hormonal, metabolic, neurologic, and immunologic processes.

In the study by Pradas et al. [66] both hematological and biochemical parameters were analyzed in elite padel players through blood sample extractions before and after a simulated competition. Authors found that men showed higher baseline values (*p* < 0.05) in red blood cells, hematocrit, hemoglobin, urea, creatinine, uric acid, albumin, glutamic oxaloacetic transaminase (GOT), glutamic-pyruvic transaminase (GPT), lactate dehydrogenase (LDH), and creatine kinase (CK) than women. When attending to the match effect, significant differences (*p* < 0.05) were obtained in urea, creatinine, CK, glucose, sodium, and magnesium. Finally, the group x match interaction revealed significant differences in serum concentrations of sodium and chloride (*p* < 0.05). They attached these gender differences to the higher intensity and anthropometric characteristics of male padel players. Equally, the game intensity could be the cause of muscle damage, protein catabolism, and electrolyte loss, where recovery and fluid intake strategies could play an important role in enhancing the training quality and performance.

### 3.3. Biomechanics

Granda-Vera et al. [52] analyzed, through a kinematic analysis of national and international padel players, the existence of visual signals (pre-cues) in drive strokes made from the back of the court after the ball hits a wall. They found, in right-handed players, that the hand height and the position of the left heel (r = 0.896 *p* < 0.001 and r = 0.777 *p* < 0.001) at the beginning and at the end of the movement were strongly related to the ball direction through the opposite court. This means a right-handed player hitting the ball in a high hand and an opened left heel position, in reference to the court’s longitudinal axis, results in a shot to the left half of the court, with the striking player as reference, and vice versa for left-handed players. These findings could serve to improve perceptual anticipation ability at all playing levels.

The foot role during padel play was analyzed by Priego-Quesada et al. [9] in order to prevent foot and ankle joint damage caused by padel movements. They found that the forefoot area supports the greatest stresses during lateral shifts, forward movements, split-steps, and pivot turns, which could lead to an excess plantar overload and provoke foot-related injuries such as sesamoiditis, plantar fasciitis, or stress fractures. These results highlight the importance of adequate specific padel footwear, as well as the training of the foot core, ankle joint mobility, and foot–ankle proprioception.

### 3.4. Epidemiology of Injuries

García-Fernández et al. [11] found, in their epidemiology research, an injury rate of 2.75 injuries per 1000 h of padel exposure, occurring most frequently towards the end of any given game or practice (42%). Priego-Quesada et al. [68] reported that 40% of the players had at least one injury during the past year. This data is below that confirmed by Sanchez-Alcaraz et al. [10] with a 71.6% incidence of injury.

Tendon injuries are the most common in padel players, followed by muscle and ligament/joint injuries [11,68]. As for their severity, mild injuries were related to muscle-tendinous upper body injuries (*p* < 0.05) and moderate injuries were related to lower body ligament injuries (*p* < 0.05). Most injuries were without any contact (68%), with muscle overload as the most common reason [11].

Regarding injury locations, lower limbs seemed to be the most common location among padel players [10,68] with a close relationship to muscle overloading (*p* < 0.001) and recurrent injuries (*p* < 0.05) [11]. Other frequent injury locations, such as the lower back, knee, shoulder, hamstrings, calf, and plantar fascia have been reported [11,32,33,34,68].

Castillo-Lozano et al. [32,33,34] observed that age, body mass index, and laterality (*p* < 0.05) were injury incidence determinants which could explain between 7.5% and 68.5% of injuries. These results were similar to those found by Garcia-Fernandez et al. [11] who reported a higher injury rate in relation to age (*p* < 0.05) and IMC increase (*p* < 0.01). Age also appears to be related to the type and incidence of injuries. Sanchez-Alcaraz et al. [10] found a higher rate of muscular lesions in players older than 35 years (23.2%) and tendinosis in players younger than 35 years (17.2%) (*p* < 0.01).

### 3.5. Match Analysis

According to Sánchez-Alcaraz 2020 [80], approximately 70% of professional padel matches are resolved in two sets. Considering the players’ gender, a significantly higher number of balanced sets and tie-break sets were observed in the male category.

#### 3.5.1. Temporal Analysis

The time of play and break time per rally have been reported in high level players as being 12.70 ± 10.05 s and 14.95 ± 6.32 s, respectively. The total time of play corresponded to 45.92% of the total time of the match, with a work:rest ratio of 0.84 [60]. The rallies’ duration distribution is commonly between 3 to 6 s (23.2%), 6 to 9 s (29.3%), and 9 to 12 s (19.6%) [6]. Break times and number of breaks seems to be influenced both by the duration of the set (*p* < 0.05) [60], the importance of the point (*p* < 0,001), and the use of “no-ad” scoring (*p* = 0.007) [86]. Regarding game duration during a tournament, Sanchez-Alcaraz et al. [86] reported higher durations (*p* = 0.004) at the semi-finals. No differences between male and female matches were found [81]

Gender differences have been reported according to temporal variables in padel games, with higher values in female than in male matches concerning the duration of rallies, number of strokes per rally, real play time, resting time, resting time per rally, and rallies per match [6,49,50,55]. Nevertheless, García-Benítez et al. [49] reported a higher effective playing time (%), more games per set, and a longer rally duration in U-18 players (*p* < 0.01). Sex similarities in work:rest ratio were found.

According to age, a shorter game duration (*p* < 0.05), rally duration (*p* < 0.01) and rest interval time between rallies (*p* < 0.01) were found in U-16 male players compared with U-18 players. In females, shorter rest interval times between rallies in U-18 players was observed (*p* < 0.01) [49].

When comparing different levels, a higher rate of play (shots per second) was found in national players compared to regional players (*p* < 0.05). As for recreational players (*p* < 0.001), they differed from national and regional players in the rally time, number of shots per point, distance covered, rate of play, and speed during active play [71].

#### 3.5.2. Technique Analysis

Volley, direct smash, and backhand strokes have been reported to be the most common types of strokes among players [6,40,57,69]. Lobs seems to be the most common shot to approach the net. In fact, performing deep lobs increased the likelihood of the continuity of rallies (*p* = 0.004) [59,61,73] as the tray does, with a percentage of point continuity of almost 90% [15].

In order to win the rally, an advantage for the serving pair has been reported [14,73]. To this aim, the smash followed by a forehand drop shot, forehand and backhand volleys, and the tray shot seem to be the most successful shots for winning points [57]. Besides, comparing winners to losers, winners made a higher percentage of smashes and trays and a lower number of side-wall shots, side and back wall, and wall boast than the losers (*p* < 0.001) [16]. Furthermore, the shot effectiveness seems to be a key factor in professional padel that distinguishes between winning and losing players (*p* < 0.05) [48].

Shots are also determined by the game style defined by the court zone: net, middle, and baseline. Some authors have found that in the net (offense) stood the use of volleys, trays and smashes, where the last were more likely to fail by an unforced error [44]. Conversely, in the baseline (defense), the use of corner side walls, groundstrokes, and lobs were shown to be relevant. The middle game (transition) was characterized by a greater use of a backhand volley in the center, direct, and tray in the sides and smash to solve the point [40,85].

Hand dominance and gender have been reported to also affect strokes, where those who are right-handed made more lobs and crossed shots. Otherwise, those who are left-handed made more directs [38,78]. Comparing hand dominance effectiveness, those who are left-handed scored more points using smashes, made more errors using the wall, and used direct shots to continue the rally. On the other hand, those who are right-handed scored more points using wall groundstrokes, with larger errors in volleys, and used more lobs to continue the rally (*p* < 0.05) [38]. Regarding gender, a higher number of strokes per rally, number of first service faults, percentage of strokes from the midfield zone, winners and errors per game, breakpoints, and lobs per match in female games, compared to male games, have been reported [47,50,55]. Conversely, higher values of percentages of the backhand lob, backhand volley, indirect forehand lob, and strokes close to the net and first service faults (*p* < 0.05) were found in male matches [6,55]. Furthermore, males obtained a higher percentage of successful first serves (*p* < 0.05) and won a higher percentage of points in a serve situation (*p* < 0.01) than females [14].

Finally, the stroke distribution seemed to be also affected by age. García-Benítez et al. [49] reported less strokes and lobs per rally in U-16 than U-18 male players (*p* < 0.01). Contrary to this, U-16 female players made more lobs per rally than U-18 (*p* = 0.01).

#### 3.5.3. Movements and Positioning Dynamics 

Movements and positioning dynamics have been tracked by some authors [77]. Their results showed a significant relationship between offensive player movements (forward, backward, or sideways) according to the position of the ball in defensive zones. Regarding players’ width positioning, when the ball is in one corner of the defensive zone, both offensive players move to that side of the court. However, if one offensive player moves close to the net, his partner moves some meters back. Regarding tactic formations, Ramón-Llin [75] found how high level players used a significantly higher percentage of the Australian formation than beginners, even though it has been reported that serving pairs won a higher percentage of points using traditional tactics instead of the Australian formation [74].

## 4. Discussion

### 4.1. Summary of Evidence 

The augment of research regarding padel in the last decade has contributed to a better understanding of the sport and has helped us to characterize the main topics of interest and detect less researched areas in the scientific literature. This scoping review reveals that match analyses were the primary focus of the included reports, covering more than half of the total. This wider and increasing knowledge in the last two years included timing, strokes, movements, positioning, and differences between genders and playing levels is probably due to their strong influence on the match outcome.

Physiology, physical, and mental performances were other important topics of interest in which authors were focused, comprehending about 25% of the articles included in our research. On the one side, players’ VO_2max_, heart rates, and thresholds were determined in order to quantify the cardio-respiratory demands of the sport [12,28,31,53,58]. On the other hand, biochemical and hematological parameter research showed that padel causes muscle damage and protein catabolism [66], which could be taken into account both for coaches and athletes to plan better recovery strategies. Apart from that, some authors reported physical performance tests involving strength, power, and the ability to repeat sprints [8,42,58,65,67,87], serving as a reference for those athletes aiming to improve their performance. Another interesting—and less researched—issue was mental performance, with only one report focused on it, suggesting that inter-game mental fatigue could impair reaction times [45].

The anthropometric profile, epidemiology of injuries, and biomechanics comprised less than a quarter of all the reviewed studies. Authors showed that higher level players presented a predominant meso-endomorphic profile, while lower level players had a predominantly endomorphic profile [7,8,35,56,62,63]—something which could serve as a reference to build a nutritional plan for the athletes. The main interest of injury-focused reports was the incidence of injury, its severity, the locations, and the possible causes, giving a quite complete overview of the sport epidemiology. Lower limbs—plantar fascia, the calf, knee, and hamstrings, the lower back, and the shoulder are the most common injury locations [11,32,33,34,68] and an injury-prevention intervention should be taken into account to avoid them and achieve player consistencies among the season. Finally, the only two biomechanics-focused studies included reports that had completely different approaches. On the one hand, an injury-related study [9] highlighted the importance of adequate specific padel footwear, as well as the training of the foot core, ankle joint mobility, and foot–ankle proprioception. On the other hand, a kinematic analysis was used in order to find pre-cues in certain drive strokes, suggesting the importance of improving the perceptual anticipation ability in players [52].

Once we observed all the scientific knowledge regarding padel, we identified some gaps, such as mental performance and fatigue, which is an unresearched field which could lead to a better understanding of how players deal with match outcomes, unforced errors, and several matches in a short period of time. In the field of injuries, the effectiveness, or not, of an injury-prevention plan during a given period would be a powerful tool for both coaches and players in order to avoid injuries and lengthen players careers. Last but not least, deeper research on how visual signals and anticipation affects match outcomes could play an important role at all playing levels.

### 4.2. Limitations 

A limitation was the inability to access certain records whose abstracts suggested they would be highly relevant. Another limitation of this review was the decision not to include books, book chapters, or literature in languages other than English. The identification of many books and book chapters addressing interventions for eco-anxiety during the selection process suggests that the inclusion of these materials in future reviews could provide rich insight.

### 4.3. Conclusions

Padel is an emerging sport both in sport and research terms. This scoping review could serve as a point of reference for coaches and researchers, with the ultimate goal of making athletes better by showing all the knowledge available in the five studied fields of interest—the anthropometric profile, physiology and physical performance, biomechanics, epidemiology of injuries, and match analyses. Furthermore, this study enables us to make a map of the current state of the research about padel and opens up doors to future investigations regarding those possible gaps in knowledge, mentioned before, such as mental performance, injury prevention interventions, and the role of anticipation and visual signals on match outcomes.

## Figures and Tables

**Figure 1 ijerph-19-04395-f001:**
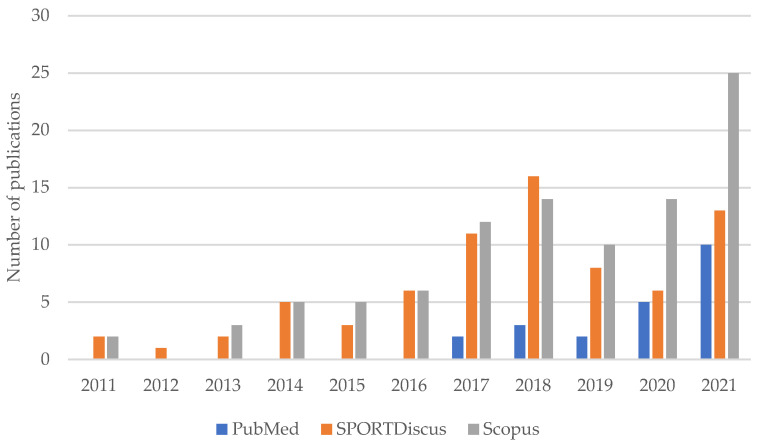
Articles containing the terms “padel” or “paddle tennis” in their title and/or abstract referring to the sport published in the databases PubMed, SPORTDiscus, and Scopus in the last 10 years (self-made figure).

**Figure 2 ijerph-19-04395-f002:**
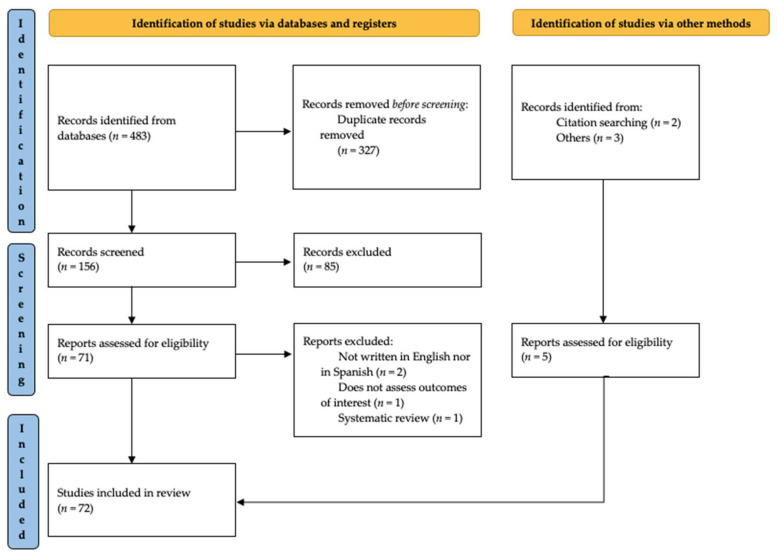
Workflow adapted from the PRISMA statement [27] showing the method to identify and select full-text articles for eligibility.

**Table 1 ijerph-19-04395-t001:** Studies included in the review, authors, year of publication, areas of intervention, number, gender, and level of the subjects.

Authors	Year	Areas of Intervention	*n*	Level
Amieba et al. [28]	2013	Physiology and physical performance	8 males	Amateur
Bartolomé et al. [29]	2016	Physiology and physical performance	16 males	Regional
Carbonell et al. [30]	2017	Physiology and physical performance	9 females	Amateur
Carrasco et al. [31]	2011	Physiology and physical performance	12 males	Top-level
Castillo-Lozano, R. [32]	2017	Epidemiology of injuries	107 males 24 females	Amateur Amateur
Castillo-Lozano, R. [33]	2015	Epidemiology of injuries	54 males 6 females	Amateur Amateur
Castillo-Lozano, R. [34]	2017	Epidemiology of injuries	54 males 59 females	Amateur Amateur
Castillo-Rodríguez et al. [12]	2014	Physiology and physical performance	24 males	National
Castillo-Rodríguez et al. [35]	2014	Anthropometric profile	36 males 12 females	Elite Elite
Courel-Ibáñez et al. [36]	2017	Match analysis	10 males	Elite
Courel-Ibáñez et al. [37]	2015	Match analysis	15 males	Elite
Courel-Ibáñez et al. [38]	2018	Match analysis	4 males	Elite
Courel-Ibáñez et al. [39]	2017	Match analysis	16 males	Elite
Courel-Ibáñez et al. [40]	2019	Match analysis	4 males	Elite
Courel-Ibáñez et al. [41]	2018	Physiology and physical performance	60 females	Amateur
Courel-Ibáñez et al. [42]	2020	Physiology and physical performance	18 males	Amateur
Courel-Ibáñez et al. [43]	2021	Physiology and physical performance	19 males 15 females	Amateur Amateur
Courel-Ibáñez, J. [44]	2021	Match analysis	# males	Elite
Díaz-García et al. [45]	2021	Physiology and physical performance	9 males 5 females	Elite Elite
Díaz-García et al. [46]	2017	Physiology and physical performance	8 males	Amateur
Escudero-Tena et al. [13]	2020	Match analysis	# females	Elite
Escudero-Tena et al. [47]	2021	Match analysis	# males # females	Elite Elite
Escudero-Tena et al. [48]	2021	Match analysis	# males # females	Elite Elite
García-Benítez et al. [49]	2017	Match analysis	16 males 16 females	National National
García-Benítez et al. [50]	2016	Match analysis	18 males 10 females	Elite Elite
García-Fernández et al. [11]	2019	Epidemiology of injuries	332 males 146 females	Regional Regional
García-González et al. [51]	2015	Epidemiology of injuries	1172 males 444 females	Amateur Amateur
Granda-Vera et al. [52]	2019	Biomechanics	4 males 1 female	Elite Elite
Hoyo et al. [53]	2007	Physiology and physical performance	12 males	National
Lozano-Sánchez et al. [54]	2020	Epidemiology of injuries	1 male	Amateur
Lupo et al. [55]	2018	Match analysis	12 males 10 females	Elite Elite
Martínez-Rodríguez et al. [56]	2015	Anthropometric profile	21 males	National
Mellado-Arbelo et al. [57]	2019	Match analysis	20 males	Elite
Müller et al. [58]	2018	Physiology and physical performance	21 males 14 females	Regional Regional
Muñoz et al. [59]	2016	Match analysis	# males	Elite
Muñoz et al. [60]	2016	Match analysis	# males	Elite Regional
Muñoz et al. [7]	2021	Anthropometric profile	40 males	Regional
Muñoz, D. [61]	2017	Match analysis	# males	Regional
Pradas et al. [62]	2019	Anthropometric profile	15 males 14 females	Elite Elite
Pradas et al. [63]	2014	Anthropometric profile, physiology, and physical performance	6 females	Elite
Pradas et al. [64]	2021	Physiology and physical performance	10 males 14 females	Elite Elite
Pradas et al. [65]	2021	Physiology and physical performance	13 males 15 females	Elite Elite
Pradas et al. [66]	2020	Physiology and physical performance	14 males 16 females	Elite Elite
Pradas et al. [67]	2021	Physiology and physical performance	15 males 15 females	Elite Elite
Priego-Quesada et al. [68]	2018	Epidemiology of injuries	46 males 34 females	Amateur Amateur
Priego-Quesada et al. [69]	2013	Match analysis	20 males	Elite
Priego-Quesada et al. [9]	2014	Biomechanics	10 males	Amateur
Ramón-Llin et al. [16]	2020	Match analysis	24 males	National
Ramón-Llin et al. [70]	2018	Match analysis	7 males 7 males	Elite National
Ramón-Llin et al. [71]	2017	Match analysis	20 males 20 males 20 males	National Regional Amateur
Ramón-Llin et al. [72]	2013	Match analysis	# males # males # males	National Regional Amateur
Ramón-Llin et al. [73]	2019	Match analysis	26 males	Elite
Ramón-Llin et al. [74]	2021	Match analysis	36 males	National
Ramón-Llin et al. [75]	2021	Match analysis	36 males 36 males	National Regional
Ramón-Llin et al. [76]	2020	Match analysis	36 males 36 males	National Regional
Ramón-Llin et al. [77]	2021	Match analysis	24 males	Elite
Ramón-Llin et al. [78]	2021	Match analysis	16 males	National
Rivilla-García et al. [79]	2019	Match analysis	14 males 30 males	National Regional
Sánchez-Alcaraz et al. [10]	2019	Epidemiology of injuries	75 males 73 females	Regional Regional
Sánchez-Alcaraz et al. [14]	2020	Match analysis	7 males 7 females	Elite Elite
Sánchez-Alcaraz et al. [80]	2020	Match analysis	# males # females	Elite Elite
Sánchez-Alcaraz et al. [81]	2021	Match analysis	12 males 12 females	Elite Elite
Sánchez-Alcaraz et al. [82]	2016	Match analysis	12 males	Regional
Sánchez-Alcaraz et al. [83]	2019	Match analysis	# males	Regional
Sánchez-Alcaraz et al. [84]	2020	Match analysis	# males	Elite
Sánchez-Alcaraz et al. [15]	2020	Match analysis	10 males 10 females	Elite Elite
Sánchez-Alcaraz et al. [85]	2021	Match analysis	48 males	Regional
Sánchez-Alcaraz et al. [86]	2020	Match analysis	# males # females	Elite Elite
Sánchez-Alcaraz et al. [87]	2018	Physiology and physical performance	8 males 9 females	Amateur Amateur
Sánchez-Alcaraz, B. [88]	2014	Physiology and physical performance	16 males	Regional
Sánchez-Muñoz et al. [8]	2020	Anthropometric profile, physiology, and physical performance	25 males 35 females	Elite Subelite
Torres-Luque et al. [6]	2015	Match analysis	8 males 8 females	Elite Elite

# Number of subjects not specified.

## Data Availability

Not applicable.

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
