# Peer review of "Performance Outcome Measures in Padel: A Scoping Review"

_ijerph, 2022, doi:10.3390/ijerph19074395_

Round 1
Reviewer 1 Report
Dear Authors,
In this article, you proposed a scoping review for performance outcome measures. The article is of scientific interest, but there are some issues that have to been addressed.
Please include the PRISMA-ScR checklist.
I suggest a deep grammar check.
Introduction
“In padel, two pairs confront each other following tennis rules and scoring system”. What do you mean following the tennis rules? In tennis there are no synthetic glass and metal court. Please rephrase.
“In padel, two pairs confront each other following tennis rules and scoring system but it is played in an enclosed synthetic glass and metal court allowing the ball to bounce on lateral and back walls for rallies.” Please provide a reference.
“The court measures 20x10 m (length x wide) and it is divided by a normal tennis net (0.88 m at the centre strap and 0.92 m at the post) in the middle.” Please provide a reference.
“These characteristics lengthen rallies so the number of actions and strokes per player is higher in comparison to similar racket sports like badminton, tennis or squash” Please explain why these characteristics lengthen rallies.
“In the last years, an augment in scientific research around padel have arisen to a bet-41 ter understanding of its characteristics and requirements both for professional and non-42 professional players.” Please provide a reference
Methods
“Study selection was performed by two independent reviewers, which would avoid to abusively eliminate an article, based on the abstracts and keywords.” How did you resolve disagreements on study selection and data extraction?
“The initial search returned 483 studies in database searching and five studies via other methods”. Please, indicate the other methods.
“ Regarding the six articles related to padel anthropometrics”. Please reference them
“3.4. Injuries”. I suggest to change Injuries in Epidemiology of Injuries.
“3.5.2. Stroke analysis”. I suggest to change Stroke analysis in Technique analysis.
Discussion
One of the aims of a scoping review is to identify gaps in the literature to aid the planning and commissioning future research. You addressed it only marginally. Please improve this aspect in discussion.
Figure 2 and 3 are redundant
Reviewer 2 Report
It is a topic that provides evidence to a sport that is currently in high demand. I think there are big mistakes that should be corrected - The title must include "systematic review" - The inclusion criteria should not be the same as the exclusion criteria - An assessment of bias has not been included in the methodology - In results table 1 is unnecessary. -The conclusions are general, I recommend more specific.Author Response
Please see the attachment

Round 2
Reviewer 1 Report
thank you for following my suggestions.
I think that the manuscript has been sufficiently improved.
About reference [5], I suggest to provide the reference after each sentence (full stop).
Author Response
Dear Reviewer,
We would like to thank you for the final appreciations about the article so we can improve its quality.
See them answered below highlighted in blue.
Introduction
About reference [5], I suggest to provide the reference after each sentence (full stop).
From lines 30 to 40, reference [5] has been included at the end of each sentence, with the text appearing as follows:
In padel, two pairs of players confront each other following tennis scoring system and similar rules regarding timing, position of players, sides, service, returns and scoring, with the difference that it is played in an enclosed synthetic glass and metal court allowing the ball to bounce on lateral and back walls for rallies [5]. The court measures 20x10 m (length x wide) and it is divided by a normal tennis net (0.88 m at the centre strap and 0.92 m at the post) in the middle [5]. The back wall is 3x10 m (height x length) and the side walls are 3x2 m ending on another 2x2 m wall [5]. The rest of the court, for each half and later side, consists of two metallic panels of equal dimension (3x2.59 m) and one gate (2x0.82 m) [5]. Each half court is composed by two service boxes, defined by the service line, which is parallel, and 6.95 m distant with respect to net and a perpendicular line dividing the service and net lines [5].

Reviewer 2 Report
Congratulations on the study and the changes made.Author Response
Thank you very much for all your comments and appreciations about the paper.